# Detection of soil-transmitted helminths and *Schistosoma* spp. by nucleic acid amplification test: Results of the first 5 years of the only international external quality assessment scheme

**Annemiek H. J. Schutte[1], Rob Koelewijn[1], Sitara S. R. Ajjampur[2], Bruno Levecke[3], James S. McCarthy[4], Rojelio Mejia[5], Steven A. Williams[6,7], Jaco J. Verweij[8], Lisette van Lieshout[9], Jaap J. van Hellemond[1] ***

1 Department of Medical Microbiology and Infectious Diseases, Erasmus MC University Medical Center, Rotterdam, The Netherlands, 2 The Wellcome Trust Research Laboratory, Division of Gastrointestinal Sciences, Christian Medical College, Vellore, Tamil Nadu, India, 3 Department of Translational Physiology, Infectiology and Public Health, Ghent University, Merelbeke, Belgium, 4 The Peter Doherty Institute for Infection and Immunity, University of Melbourne, Australia, 5 Department of Pediatrics–Tropical Medicine, Baylor College of Medicine, Houston, Texas, United States of America, 6 Department of Biological Sciences, Smith College, Northampton, Massachusetts, United States of America, 7 Molecular and Cellular Biology Program, University of Massachusetts, Amherst, Massachusetts, United States of America, 8 Laboratory for Medical Microbiology and Immunology, Elisabeth-Tweesteden Hospital, Tilburg, the Netherlands, 9 Leiden University Center for Infectious Diseases (LU-CID); Parasitology group, Leiden University Medical Center, Leiden, The Netherlands

* j.vanhellemond@erasmusmc.nl

## Abstract

### Background

Infections with soil-transmitted helminths (STH) and schistosomiasis (SCH) result in a significant global health burden, particularly in rural communities in low and middle-income countries. While microscopy remains the primary diagnostic method for STH and SCH in resource-limited settings, nucleic acid amplification tests (NAATs) are gaining prominence as tools for evaluation of public health control programs in endemic countries, and individual diagnosis in high-income countries. Despite the high sensitivity and specificity of NAATs, previous research has highlighted inter-laboratory variations, both in technical and clinical performance, justifying the need for continuous proficiency testing.

### Methodology

Results from 5 rounds over a 5-year period of the so far only longitudinal international Helminth External Molecular Quality Assessment Scheme (HEMQAS), coordinated by the Dutch Foundation for Quality Assessment in Medical Laboratories (SKML), were examined in order to (i) assess the diagnostic proficiency of laboratories in detecting helminths in stool and (ii) identify potential factors contributing to variations in performance.

**Data Availability Statement:** All the raw data used to prepare the figures are supplied in the supporting information files.

**Funding:** The author(s) received no specific funding for this work.

**Competing interests:** The authors have declared that no competing interests exist.

## Outcome and conclusions

Thirty-six laboratories, from 18 countries and 5 continents, participated in HEMQAS. The overall diagnostic performances were satisfying, with remarkably low numbers (<2%) of false-positive results. False-negative results were more often reported for stool (15%) than for DNA (5%) samples. False-negative results varied largely between targets (the highest number (29%) for *Trichuris trichiura*). Twenty-five laboratories provided a sufficient number of results for a robust comparison between participating laboratories, which confirmed substantial inter-laboratory variability in quantitative NAAT results (Cq-values). This variability likely arises from differences in pre-treatment, DNA isolation and DNA-target amplification procedures. This study emphasizes the complexity of molecular diagnosis for STH and SCH, highlighting the critical role of proper stool preparation and DNA isolation methods. The results underscore the necessity for laboratory professionals and public health decision-makers to recognize these complexities and continuously undertake external quality assessment schemes to ensure accurate and reliable performance in molecular diagnosis.

## Author summary

Parasitic worms cause significant global health challenges, particularly in rural communities in low and middle-income countries. Laboratory tests that detect parasite DNA in human stool are increasingly being used for the diagnosis of parasitic worm infections. While parasite DNA detection can be an excellent diagnostic tool, its performance in clinical practice varies considerably, as demonstrated by previous research. To ensure the quality in diagnostic testing of these parasitic worms, it is important that laboratories evaluate the diagnostic performance of their DNA-based tests by participating in a so-called external quality assessment scheme (EQAS). This study, focusing on the first five-yearly rounds of the so far only longitudinal international EQAS to date, reveals that quantitative DNA detection results are not directly comparable across laboratories due to substantial inter-laboratory variability. This variability mainly stems from differences in pre-treatment and DNA isolation protocols, and to a lesser extent from differences in DNA target amplification procedures. Furthermore, the study underscores the complexity of molecular diagnosis for parasitic worms in clinical practice and the necessity for laboratories to engage in EQAs to guarantee the proper functioning of these diagnostic tests.

## Introduction

Infections with soil-transmitted helminths (STH) and schistosomiasis (SCH) lead to a major global health burden due to their high prevalence and intensity of infection in rural communities. STH are caused by hookworms (*Necator americanus*, *Ancylostoma duodenale* and *Ancylostoma ceylanicum)*, whipworms (*Trichuris trichiura*), the roundworm (*Ascaris lumbricoides*) and *Strongyloides stercoralis*. Globally, over 1 billion people are estimated to be affected by one or multiple STH, and circa 140 million people by SCH [1–3]. In many high-prevalence countries, public health programs entail large-scale deworming programs aiming to control and eventually eliminate STH and SCH.

In 2020 the World Health Organization (WHO) released the 2030 targets for global advancements in STH control and among the listed requirements is the need for improved

and less time-consuming diagnostics [4,5]. In resource-poor settings microscopy-based techniques are the primary method for diagnosing both STH and SCH. However, nucleic acid amplification tests (NAATs) are increasingly available and used for research purposes in public health programs in endemic countries as well as in individual testing in high-income countries [6–8]. Compared to microscopy, NAATs have a very high diagnostic accuracy, both in specificity and sensitivity, and can therefore be used to accurately assess changes in infection prevalence following mass drug administration [9–13]. Furthermore, NAATs may be less labor-intensive, especially if large numbers of samples need to be examined in a high-throughput setting [14]. Lastly, NAATs may play a role in the early detection of emerging anthelmintic resistance [15].

Although the detection of STH and SCH by NAATs can be both sensitive and specific, technical difficulties are recognized as possible impediments to diagnostic performance [8,16,17]. First, DNA amplification can only be achieved if target DNA is released from eggs or larvae; the former are surrounded by a firm eggshell or cuticle that varies in structure and thickness between helminth species [17]. Second, the quantity of DNA within the egg and larvae can be dependent on the development of the embryonic stage [17]. Further, stool is known to contain factors that inhibit DNA amplification, which poses difficulties for the proper amplification of DNA extracted from stool [18]. In a world that is increasingly dependent and trusting towards NAATs, it is crucial that the NAAT performance in actual practice is secured. External quality assessment schemes (EQAS) provide a blinded process in which well-validated specimens can be examined and results can be compared with other diagnostic laboratories. Thereby EQAS are essential for the detection of weaknesses in routine diagnostics that can be improved [19,20].

In 2018, the Helminth External Molecular Quality Assessment Scheme (HEMQAS) was launched as the first international EQAS for the detection of STH and SCH in stool by NAATs. It demonstrated proof of principle for the feasibility of a worldwide EQAS for NAATs for detection of helminths in stool [21]. It also demonstrated the added value of the simultaneous distribution of both stool and DNA samples in HEMQAS, as this combination allows the assessment of accuracy of the entire laboratory procedure as well as the efficiency of target DNA amplification. Thereby each HEMQAS participant can determine their areas for improvement, such as DNA isolation or amplification of the target DNA.

In this study all results of the HEMQAS of the first 5-years are examined with the following aims: (i) to evaluate the diagnostic performance of laboratories for the detection of different helminths, and (ii) to identify potential causes for the differences in performance between the laboratories.

## Methods

*HEMQAS design* HEMQAS is organized by the Dutch Foundation for Quality Assessment in Medical Laboratories (SKML) [22]. From 2018 to 2022, a set of 8 to 12 homogenized ethanol-preserved stool samples and 6 to 10 purified DNA samples in stabilization buffer, were sent from the SKML-EQAS coordinating center (Department of Medical Microbiology and Infectious Diseases, Erasmus Medical Center, Rotterdam, the Netherlands) to the participating laboratories. Between the different distributions the DNA and parasite loads intentionally varied with emphasis on loads challenging to detect. The participating laboratories were asked to report the detected helminths species with a corresponding Cq-value. Concurrently, all participating laboratories received an online questionnaire in which they were asked to specify the DNA extraction and NAAT protocol that was used for their NAAT.

## Sample preparation and validation

The applied methods for sample preparation, selection of expert laboratories and sample validation, were in detail described before [21]. In short, the distributed stool samples were prepared from clinical specimens that contained a high number of eggs and/or larvae as demonstrated by microscopy or PCR. The coordinating center of HEMQAS prepared homogeneous and stable suspensions of these clinical specimens by dilution in 70% (v/v) ethanol. The samples were sent only once per year for logistical and economic reasons.

Prior to each distribution the following seven expert laboratories examined selected stool and DNA samples to certify the presence or absence the 6 target helminths and to validate the homogeneity of the samples: Baylor College of Medicine, Houston, TX, U.S.A.; Christian Medical College, Vellore, Tamil Nadu, India; Elisabeth-Tweesteden Hospital, Tilburg, the Netherlands; Erasmus MC University Medical Center, Rotterdam, The Netherlands; Leiden University Medical Center, Leiden, The Netherlands; Peter Doherty Institute for Infection and Immunity, Melbourne, Australia; Smith College, Northampton, MA, U.S.A.. A sample was classified as positive for a particular helminth target if the following three criteria were fulfilled: (i) all quintuplicates analyzed by one of the expert laboratories were positive for that target, (ii) the standard deviation of the quantification cycle (Cq)-values of the quintuplicates was less than two cycles, and (iii) all other expert laboratories confirmed the presence of that target. These criteria guarantee that positive certified helminth targets should be detectable by state-of-the-art NAAT methods. A sample was classified as negative for a particular target if the two following criteria were fulfilled: (i) all quintuplicates analyzed by the expert laboratory were negative for that target and (ii) all other expert laboratories confirmed the negative result for that particular target. Samples that were not classified as positive or negative for a specific target, were considered as 'educational' for that target, meaning that the target was not consistently detected by all expert laboratories. In this case, the sample probably contained a too-low parasite load to homogeneously be dispersed and/or to be detected. Hence, a positive NAAT result cannot be demanded from the participating laboratories, and therefore, these targets are validated as 'educational'. The criteria used for validation of targets as positive, negative or educational, are commonly applied for validation of samples by EQAS organizations, including but not limited to the SKML.

## Inclusion of HEMQAS participants and data analysis

HEMQAS participation was open to all laboratories from all over the world. All participating laboratories reported their results using the online QBase submission software of the SKML. Anonymized data was extracted from Qbase, after which data were analyzed in Excel 2016 and figures were prepared using Graphpad Prism 9.1.4.

Since only 3 stool samples were distributed to the participating laboratories that contained *Ancylostoma spp*. (**Table 1**), too few results were obtained for a reliable performance analysis for this target. Therefore, this helminth target was excluded from further analysis in this study. In addition to analysis of the qualitative performance (absence / presence of target), this study also aimed to evaluate to accuracy of qualitative test results in relation to the semi-quantitative results and the used methodology. For this part of the study only the results of laboratories were included that reported a minimum 4 quantitative results for both DNA- as well as stool samples for at least one helminth target, to avoid bias because of too little data points.

# Results

## Analysis of qualitative results

Over the studied 5-year period from 2018 to 2023, 36 laboratories participated in the HEMQAS from the following countries: Australia (n = 2), Belgium (n = 1), Denmark (n = 1),

**Table 1. Overview of distributed samples and results of participants.**

| Helminth target | | *Ancylostoma spp.* | | *Ascaris* | | *Necator* | | *Schistosoma* | | *Strongyloides* | | *Trichuris* | | *Total* | |
|---|---|---|---|---|---|---|---|---|---|---|---|---|---|---|---|
| **Distributed samples** | | Stool | DNA | Stool | DNA | Stool | DNA | Stool | DNA | Stool | DNA | Stool | DNA | Stool | DNA |
| **Positive** | | | | | | | | | | | | | | | |
| | Samples (n) | 2 | 0 | 13 | 16 | 15 | 10 | 11 | 10 | 10 | 9 | 6 | 8 | 57 | 53 |
| | Participant results (n) | 27 | n.a. | 166 | 210 | 199 | 131 | 148 | 140 | 202 | 173 | 77 | 97 | 819 | 751 |
| | Positive result | 21 | n.a. | 143 | 193 | 172 | 121 | 131 | 134 | 174 | 166 | 55 | 96 | 696 | 710 |
| | False negative result | 6 | n.a. | 23 | 17 | 27 | 10 | 17 | 6 | 28 | 7 | 22 | 1 | 123 | 41 |
| | % false negative results | 22% | n.a. | 14% | 8% | 14% | 8% | 11% | 4% | 14% | 4% | 29% | 1% | 15% | 5% |
| | Maximal Cq-value | 39 | n.a. | 39 | 39 | 40 | 38 | 37 | 37 | 40 | 41 | 41 | 39 | | |
| | Minimal Cq-value | 26 | n.a. | 14 | 19 | 13 | 17 | 15 | 13 | 16 | 15 | 24 | 22 | | |
| | Median Cq value | 32 | n.a. | 30 | 30 | 28 | 32 | 25 | 26 | 28 | 26 | 32 | 29 | | |
| | Mean Cq value | 32 | n.a. | 29 | 30 | 28 | 30 | 25 | 25 | 28 | 25 | 32 | 29 | | |
| | SD Cq values | 3.5 | n.a. | 5.3 | 4.2 | 5.4 | 5.1 | 4.7 | 5.5 | 4.8 | 4.8 | 3.5 | 4.4 | | |
| **Negative** | | | | | | | | | | | | | | | |
| | Samples (n) | 47 | 36 | 31 | 20 | 30 | 26 | 37 | 25 | 37 | 26 | 30 | 28 | 212 | 161 |
| | Participant results (n) | 601 | 457 | 413 | 255 | 401 | 339 | 504 | 335 | 663 | 468 | 366 | 340 | 2948 | 2194 |
| | Negative results (n) | 593 | 449 | 407 | 253 | 398 | 334 | 500 | 331 | 653 | 457 | 364 | 340 | 2915 | 2164 |
| | False positive results (n) | 8 | 8 | 6 | 2 | 3 | 5 | 4 | 4 | 10 | 11 | 2 | 0 | 33 | 30 |
| | % false positive results | 1% | 2% | 1% | 1% | 1% | 1% | 1% | 1% | 2% | 2% | 1% | 0% | 1% | 1% |
| **Educational** | | | | | | | | | | | | | | | |
| | Samples (n) | 1 | 0 | 6 | 0 | 5 | 0 | 2 | 1 | 3 | 1 | 14 | 0 | 31 | 2 |
| | Participant results (n) | 13 | n.a. | 72 | n.a. | 62 | n.a. | 33 | 18 | 44 | 13 | 169 | n.a. | 393 | 31 |
| | Negative results (n) | 2 | n.a. | 32 | n.a. | 40 | n.a. | 21 | 16 | 26 | 3 | 89 | n.a. | 210 | 19 |
| | Positive results (n) | 11 | n.a. | 40 | n.a. | 22 | n.a. | 12 | 2 | 18 | 10 | 80 | n.a. | 183 | 12 |
| | % negative results | 15% | n.a. | 44% | n.a. | 65% | n.a. | 64% | 89% | 59% | 23% | 53% | n.a. | 53% | 61% |
| | % positive results | 85% | n.a. | 56% | n.a. | 35% | n.a. | 36% | 11% | 41% | 77% | 47% | n.a. | 47% | 39% |
| | Maximal Cq-value | 40 | n.a. | 41 | n.a. | 40 | n.a. | 37 | 40 | 40 | 38 | 43 | n.a. | | |
| | Minimal Cq-value | 22 | n.a. | 17 | n.a. | 22 | n.a. | 20 | 35 | 18 | 29 | 20 | n.a. | | |
| | Median Cq-value | 30 | n.a. | 33 | n.a. | 34 | n.a. | 29 | n.a. | 29 | n.a. | 33 | n.a. | | |
| | Mean Cq value | 30 | n.a. | 38 | n.a. | 33 | n.a. | 29 | n.a. | 29 | n.a. | 33 | n.a. | | |
| | SD Cq values | 5.1 | n.a. | 5.2 | n.a. | 4.8 | n.a. | 5.3 | n.a. | 5.3 | n.a. | 4.1 | n.a. | | |
| **Total number of samples (n)** | | *50* | 36 | 50 | 36 | 50 | 36 | 50 | 36 | 50 | 36 | 50 | 36 | 300 | 216 |

Germany (n = 1), France (n = 1), India (n = 1), Italy (n = 1), Kenya (n = 1), Luxembourg (n = 1), Malaysia (n = 1), Mozambique (n = 1) Netherlands (n = 10), Norway (n = 2), Spain (n = 1), Uganda (n = 2), USA (n = 5), Sweden (n = 1) and Switzerland (n = 3). Laboratories that participated in HEMQAS during the entire 5-year period, received a total of 50 stool samples and 36 DNA samples. However, most participating laboratories (n = 30) participated for fewer than 5 years, and thus examined a smaller number of samples. The distributed samples could comprise none, a single or multiple helminth species and **Table 1** shows the number distributed stool and DNA samples classified as positive, negative or educational for the different helminth species investigated in this study: *Ascaris lumbricoides*, *Necator americanus*, *Schistosoma* spp., *S. stercoralis* and *T. trichiura*. This table also shows both the number of reported results and the total number of false-negative and false-positive results reported by all participating laboratories.

The accuracy of the qualitative results of NAAT for the detection of helminth parasites in stool was overall on average ca. 85%, with a sensitivity of 86% for *A. lumbricoides*, 86% for *N.*

*americanus*, 89% for *Schistosoma* spp., 86% for *S. stercoralis* and only 71% for *T. trichiura*. As expected, the overall sensitivity for the detection of the helminth targets was higher in the DNA samples compared to stool samples (95% vs. 85%) with a sensitivity in DNA samples of 92% for *A. lumbricoides*, 92% for *N. americanus*, 96% for *Schistosoma* spp., 96% for *S. stercoralis*, and 99% for *T. trichiura* (**Table 1**). Participating laboratories reported overall more often false-negative outcomes (15%) than false-positive outcomes (1%) for stool samples (**Table 1**). In stool samples, most false-negative results were reported for *T. trichiura* (22 out of 77 stool samples, 29%) and in DNA samples for *Ascaris* ssp. (17 out of 210, 8%). The reported false results were not evenly distributed over the participating laboratories, because three participating laboratories (J, R and S) reported approximately half of all errors and were clearly poor performers in comparison the other participating laboratories (**Fig 1**).

## Analysis of quantitative results

In case of a positive test result the participating laboratories also reported the cycle at which the NAAT became positive (Cq-value). This Cq-value is a semi-qualitative measure for the amount of the target DNA and a low Cq-value means that relatively few DNA-amplification cycles were required to exceed the detection threshold and thus corresponds to relatively high concentrations of target DNA. However, the Cq -value is not only determined by the abundance of target DNA in the sample, as it is also affected by the efficiency of the NAAT. Since the distributed HEMQAS samples were validated to comprise equal amounts of helminths with only minor variations, the reported Cq-values could be used to examine the variation in detection efficiency among participating laboratories for distinct targets and sample types. To obtain robust data, only the results of laboratories were included that reported quantitative results for at least 4 DNA- and 4 stool samples for at least one helminth target. Using this inclusion criterion, the results of 11 participating laboratories had to be excluded, after which the results of 25 participating laboratories remained. The included laboratories were based in the following countries: Australia (n = 1), Belgium (n = 1), Denmark (n = 1), Germany (n = 1), India (n = 1), Italy (n = 1), Kenia (n = 1), Mozambique (n = 1) Netherlands (n = 9), Norway (n = 2), Sweden (n = 1), Switzerland (n = 1), Uganda (n = 1) and USA (n = 3)

According to the study of Cools et al. [21], for each positive target in each sample the median Cq-value of all participants was calculated. Subsequently, for each positive target in each sample for each included participant, their difference to the median of all participants could be calculated. These differences (the delta Cq to the median of all participants) was summed per participant per target per sample type, after which the delta median Cq-value per target could be compared among participants. In **Figs 1–5**, these delta Cq-values per target per participant are shown. In all panels of **Figs 1–5**, the participants are sorted identically on the X-axis, based on their overall median Cq-value for all helminth targets (from low to high Cq-value). The delta-Cq is plotted on the Y-axis and data points below the X-axis (negative values) correspond to participants that reported on average lower Cq-values than the median of all participants and thus represent participants with a relatively efficient NAAT. To show only robust delta median Cq-value data-points, only results were plotted of participants that reported ≥4 Cq-values for the depicted target. In addition, in each panel of **Figs 1–5**, the number of false-negative and false-positive outcomes are listed for each participant. Since the results for educational samples could not be assigned as false-positive nor false-negative, only the number and proportion of positive results for each target and sample type per participant are shown.

This analysis demonstrated a large variation in reported Cq-values for all helminth targets (**Figs 1–5**). For positive targets in most stool samples the difference between the lowest and

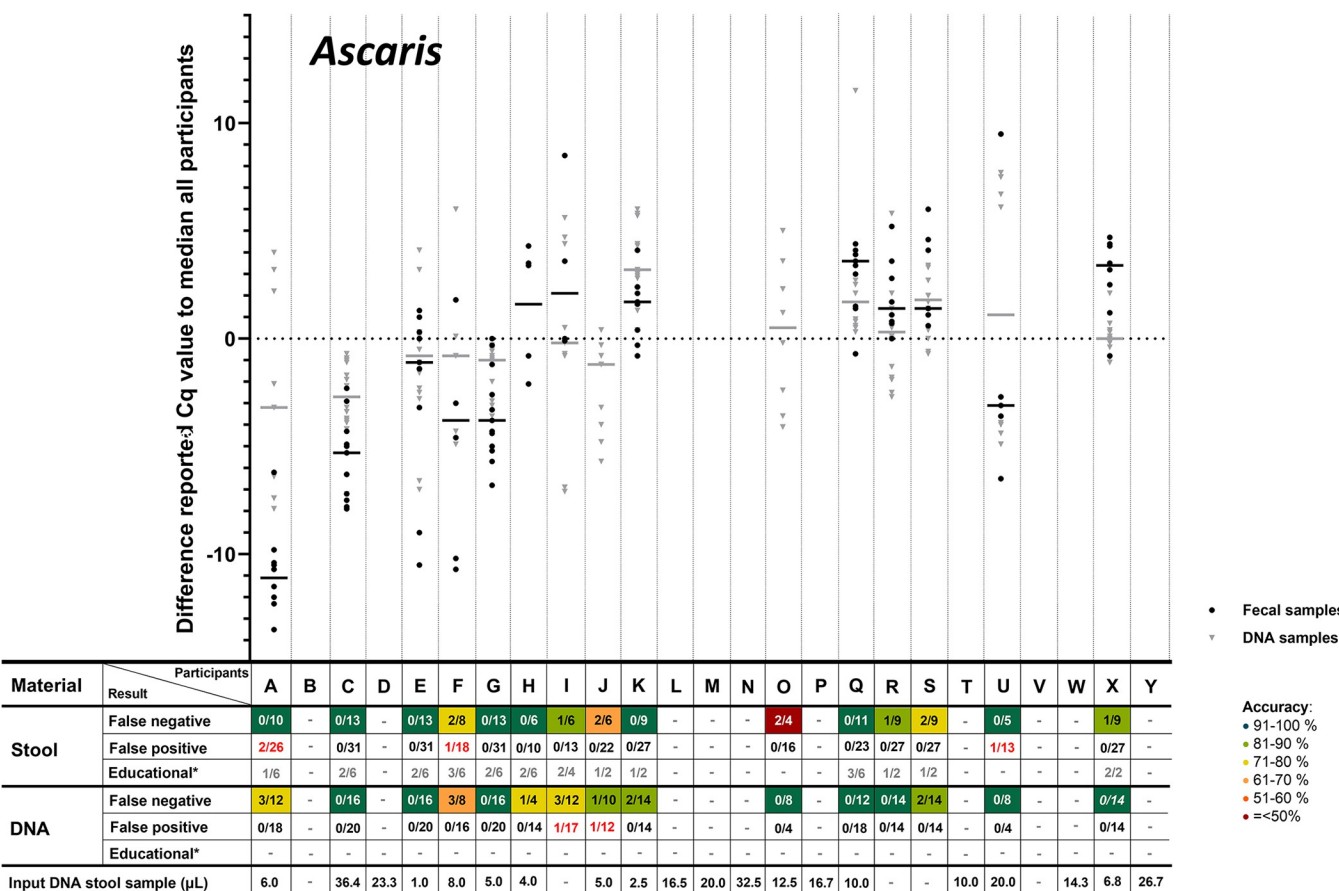

**Fig 1. Difference in reported Cq-values for stool and DNA samples positive for *Ascaris* spp.** Each point represents the reported Cq-value of the individual participant subtracted by the median of reported Cq-value of all participants and each horizontal line represents the median difference in reported Cq-value. Points and lines were only shown if the participants reported four or more Cq-values for either stool or DNA (educationals excluded). The table below represents the number of false negatives, false positives and false negative educationals (*) are displayed. Some numbers were placed in italics, this means the participant included 'positive' results without reporting a Cq-value. Each letter on the the X-axis represents a different participant, these participants are similarly ordered throughout Fig 1–5. For the target *Trichuris* (Fig 5), most samples were assigned as educationals.

highest reported Cq-value was >15 amplification cycles and for DNA samples this difference was as expected lower, but still often >10 amplification cycles. This observed variation was not randomly distributed, as some participants reported consistently relatively low Cq-values whereas others reported relatively high Cq-values. This observation is true for all helminth targets in both stool and DNA samples (**Figs 1–5**). Therefore, these results showed that large differences exist between participants in the detection efficiency of their NAAT. Participants that reported relatively low Cq-values for a specific helminth target in stool samples, tend to do so for all other helminth targets as well, because the participants are sorted identically in all panels of **Figs 1–5** and throughout all these the average reported Cq-value ascents from left to right. Thus, independent from the target, participants tended to report either a consistently low or high Cq-values for stool samples. This correlation is much less pronounced for DNA samples, which shows that sample pre-treatment before DNA isolation and the DNA isolation method itself are important factors affecting the efficiency of NAAT to detect helminths in stool.

For most helminth targets, at most a very weak correlation was observed between the incidence of reported false-negative outcomes and relatively high Cq-values (only slightly more false-negative results reported by participants on the right half of the panels of **Figs 1–5**). This suggests that the differences in efficiency between participants did not substantially affect the

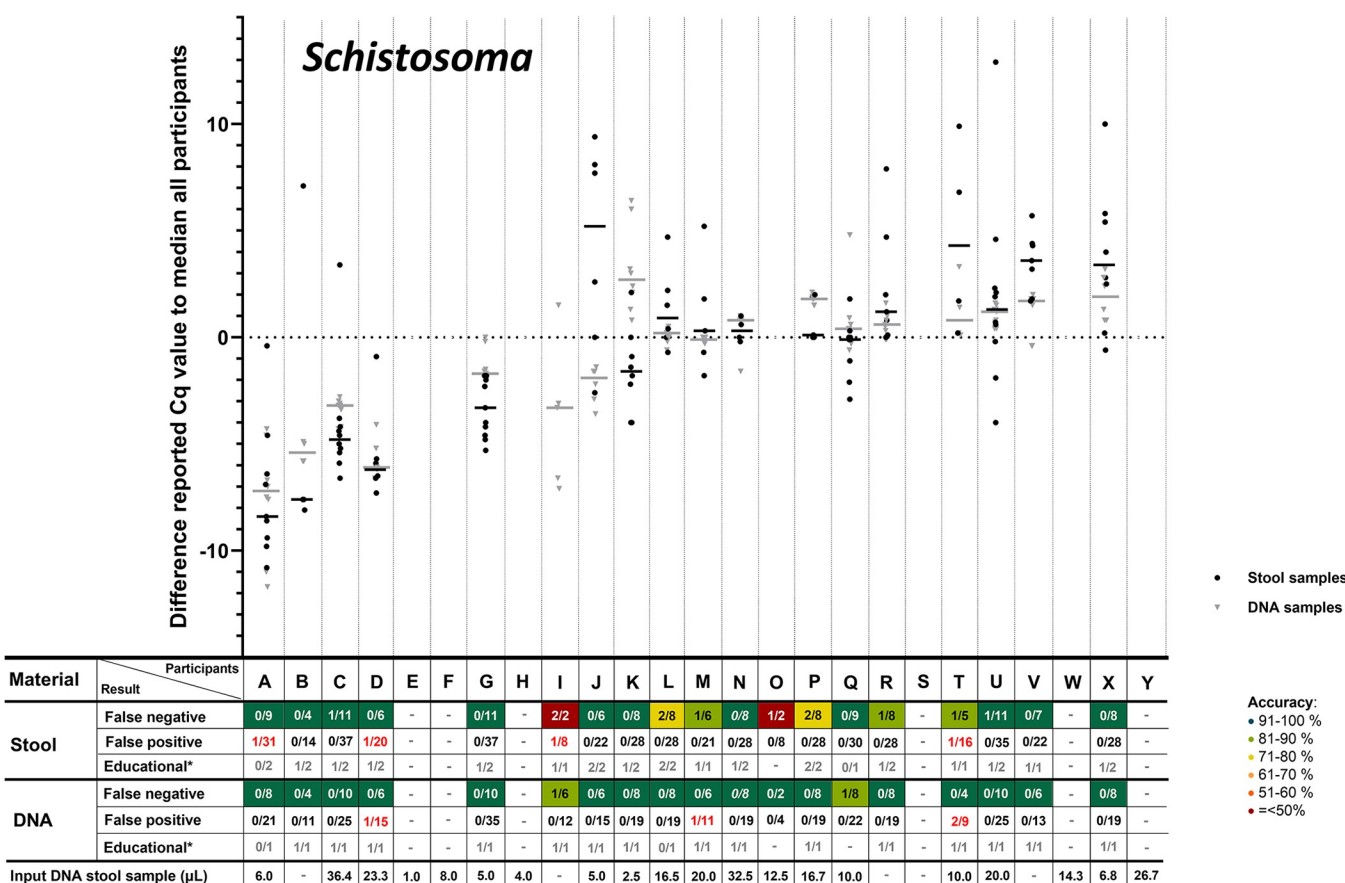

| Material | Participants / Result | A | B | C | D | E | F | G | H | I | J | K | L | M | N | O | P | Q | R | S | T | U | V | W | X | Y |
|---|---|---|---|---|---|---|---|---|---|---|---|---|---|---|---|---|---|---|---|---|---|---|---|---|---|---|
| **Stool** | False negative | 0/9 | 0/4 | 1/11 | 0/6 | - | - | 0/11 | - | 2/2 | 0/6 | 0/8 | 2/8 | 1/6 | 0/8 | 1/2 | 2/8 | 0/9 | 1/8 | - | 1/5 | 1/11 | 0/7 | - | 0/8 | - |
| | False positive | 1/31 | 0/14 | 0/37 | 1/20 | - | - | 0/37 | - | 1/8 | 0/22 | 0/28 | 0/28 | 0/21 | 0/28 | 0/8 | 0/28 | 0/30 | 0/28 | - | 1/16 | 0/35 | 0/22 | - | 0/28 | - |
| | Educational* | 0/2 | 1/2 | 1/2 | 1/2 | - | - | 1/2 | - | 1/1 | 2/2 | 1/2 | 2/2 | 1/1 | - | - | 2/2 | 0/1 | 1/2 | - | 1/1 | 1/2 | 1/1 | - | 1/2 | - |
| **DNA** | False negative | 0/8 | 0/4 | 0/10 | 0/6 | - | - | 0/10 | - | 1/6 | 0/6 | 0/8 | 0/8 | 0/6 | 0/8 | 0/2 | 0/8 | 1/8 | 0/8 | - | 0/4 | 0/10 | 0/6 | - | 0/8 | - |
| | False positive | 0/21 | 0/11 | 0/25 | 1/15 | - | - | 0/35 | - | 0/12 | 0/15 | 0/19 | 0/19 | 1/11 | 0/19 | 0/4 | 0/19 | 0/22 | 0/19 | - | 2/9 | 0/25 | 0/13 | - | 0/19 | - |
| | Educational* | 0/1 | 1/1 | 1/1 | 1/1 | - | - | 1/1 | - | 1/1 | 1/1 | 1/1 | 0/1 | 1/1 | 1/1 | - | 1/1 | - | 1/1 | - | 1/1 | 1/1 | 1/1 | - | 1/1 | - |
| | Input DNA stool sample (µL) | 6.0 | - | 36.4 | 23.3 | 1.0 | 8.0 | 5.0 | 4.0 | - | 5.0 | 2.5 | 16.5 | 20.0 | 32.5 | 12.5 | 16.7 | 10.0 | - | - | 10.0 | 20.0 | - | 14.3 | 6.8 | 26.7 |

Accuracy:
- 91-100 %
- 81-90 %
- 71-80 %
- 61-70 %
- 51-60 %
- =<50%

**Fig 2. Difference in reported Cq-values for stool and DNA samples positive for *Schistosoma* spp.** For detailed explanation of the figure, see legend of Fig 1.

qualitative outcome for stool samples with a substantial parasite load. On the other hand, participating laboratories that reported relatively low Cq-values (sorted on the left of part of the X-axis in de panels of **Figs 1–5**) seemed to report a higher proportion of educational samples as positive, an observation most apparent for *T. trichiura*. This result suggests that lower parasite loads are more often detected by efficient NAAT protocols.

## Analysis of variations in methodology

In addition to the test results, most participants also answered the additional questions on the NAAT methodology used for the detection of helminths in stool in their laboratory. The variation between laboratories in methodology was very large, encompassing differences at all stages of the NAAT procedure, including sample pre-treatment before DNA extraction, DNA extraction method and equipment, volume of stool used for DNA extraction and subsequent NAAT, DNA target for amplification, sequences of primers and probes, type of DNA probes, PCR protocol (e.g. annealing temperature and number of cycles), concentration and source of DNA polymerase and other PCR components [8, 17]. Basically each participating laboratory used a unique protocol that deviated in one or more aspects from all other participants. Therefore, no conclusions can be drawn on the correlation between variations in the various steps of the NAAT and the resulting test efficiency.

As participating laboratories reported the volume of stool used for DNA extraction, their elution volume as well as the volume of isolated DNA in their NAAT, it could be calculated

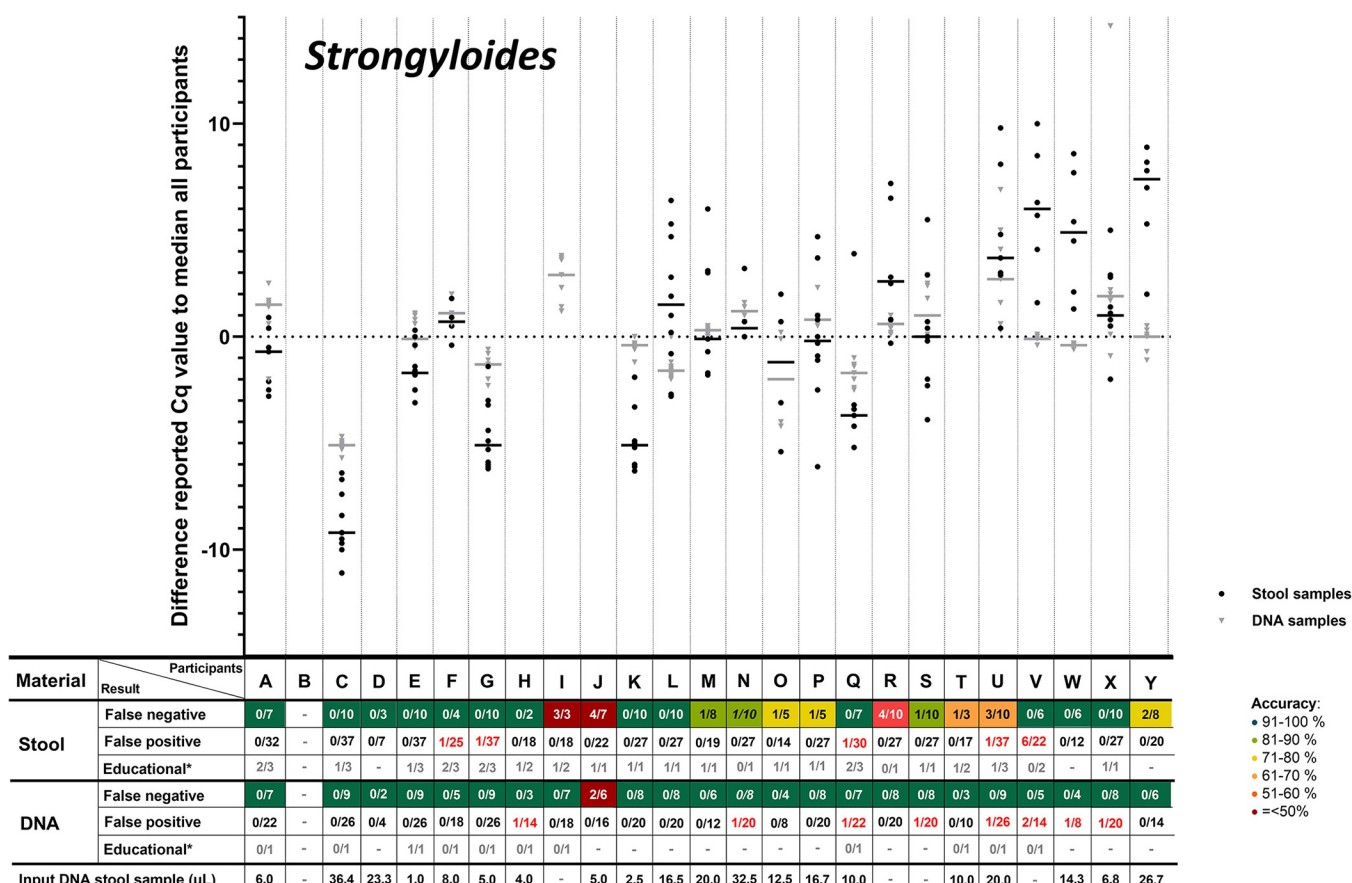

| Material | Participants / Result | A | B | C | D | E | F | G | H | I | J | K | L | M | N | O | P | Q | R | S | T | U | V | W | X | Y |
|---|---|---|---|---|---|---|---|---|---|---|---|---|---|---|---|---|---|---|---|---|---|---|---|---|---|---|
| **Stool** | False negative | 0/7 | - | 0/10 | 0/3 | 0/10 | 0/4 | 0/10 | 0/2 | 3/3 | 4/7 | 0/10 | 0/10 | 1/8 | 1/10 | 1/5 | 1/5 | 0/7 | 4/10 | 1/10 | 1/3 | 3/10 | 0/6 | 0/6 | 0/10 | 2/8 |
| | False positive | 0/32 | - | 0/37 | 0/7 | 0/37 | 1/25 | 1/37 | 0/18 | 0/18 | 0/22 | 0/27 | 0/27 | 0/19 | 0/27 | 0/14 | 0/27 | 1/30 | 0/27 | 0/27 | 0/17 | 1/37 | 6/22 | 0/12 | 0/27 | 0/20 |
| | Educational* | 2/3 | - | 1/3 | - | 1/3 | 2/3 | 2/3 | 1/2 | 1/2 | 1/1 | 1/1 | 1/1 | 1/1 | 0/1 | 1/1 | 1/1 | 2/3 | 0/1 | 1/1 | 1/2 | 1/3 | 0/2 | - | 1/1 | - |
| **DNA** | False negative | 0/7 | - | 0/9 | 0/2 | 0/9 | 0/5 | 0/9 | 0/3 | 0/7 | 2/6 | 0/8 | 0/8 | 0/6 | 0/8 | 0/4 | 0/8 | 0/7 | 0/8 | 0/8 | 0/3 | 0/9 | 0/5 | 0/4 | 0/8 | 0/6 |
| | False positive | 0/22 | - | 0/26 | 0/4 | 0/26 | 0/18 | 0/26 | 1/14 | 0/18 | 0/16 | 0/20 | 0/20 | 0/12 | 1/20 | 0/8 | 0/20 | 1/22 | 0/20 | 1/20 | 0/10 | 1/26 | 2/14 | 1/8 | 1/20 | 0/14 |
| | Educational* | 0/1 | - | 0/1 | - | 1/1 | 0/1 | 0/1 | 0/1 | 0/1 | - | - | - | - | - | - | - | 0/1 | - | - | 0/1 | 0/1 | 0/1 | - | - | - |
| **Input DNA stool sample (µL)** | | 6.0 | - | 36.4 | 23.3 | 1.0 | 8.0 | 5.0 | 4.0 | - | 5.0 | 2.5 | 16.5 | 20.0 | 32.5 | 12.5 | 16.7 | 10.0 | - | - | 10.0 | 20.0 | - | 14.3 | 6.8 | 26.7 |

**Fig 3. Difference in reported Cq-values for stool and DNA samples positive for *Strongyloides stercoralis*.** For detailed explanation of the figure, see legend of Fig 1.

what proportion of the distributed 200 µL stool was examined in their NAAT. Again, a large variation was found among participants, as some used only 1 µL of the stool sample whereas others used up to 36 µL. As this difference was large, it was examined whether a correlation could be found between this 'sample input volume for NAAT' (shown underneath all panels of **Figs 1–5**) with the median reported delta-Cq to the median Cq of all participants for each helminth target. However, no clear trend was observed (no increase or decrease in sample volume input from left to right in **Figs 1–5**). This result suggests that the 'sample input volume for NAAT' is not the critical factor that on its own can explain the observed variation in reported Cq-values. When a similar analysis was performed to examine the correlation between quantitative results for DNA samples and the NAAT methodology used, again no correlation was found (not shown). These findings demonstrate that the diversity in the methodology for DNA amplification only, was already too high to detect trends between the applied methodology and test performance. Altogether the analyses of the quantitative results show that the NAAT for detection of helminths in stool differ substantially between laboratories, and that the cause for observed variation is multi-factorial.

## Discussion

This is the first reported longitudinal inter-laboratory comparison study on the performance of NAAT for the detection of STH and SCH in stool. It demonstrated that although NAATs

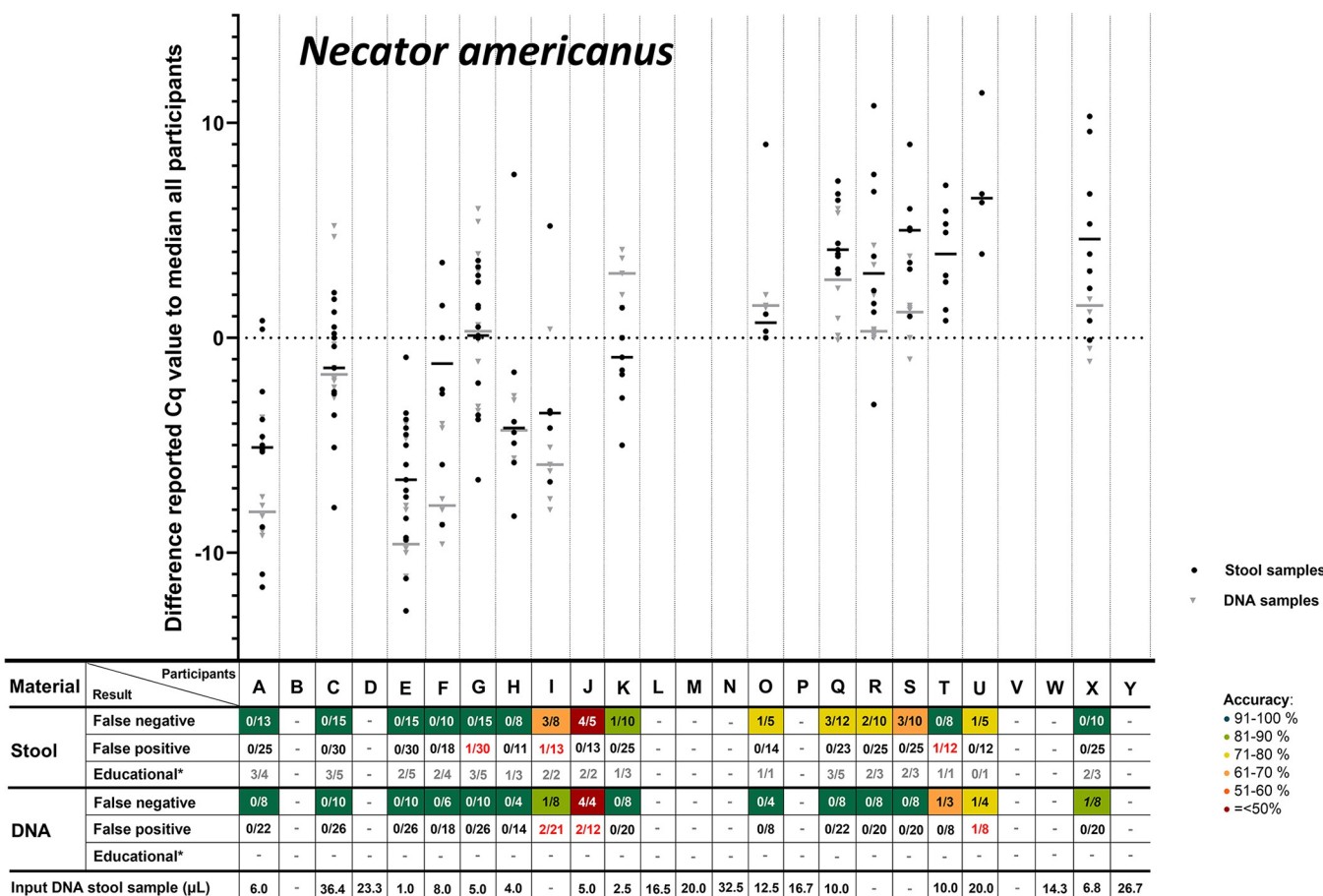

**Fig 4. Difference in reported Cq-values for stool and DNA samples positive for *Necator americanus*.** For detailed explanation of the figure, see legend of Fig 1.

can be highly sensitive and specific, their performance in routine practice for clinical or epidemiological purposes can be poor in some laboratories, as false-negative results (ca. 15%) and false-positive (ca. 1%) occur. However, a small proportion of the participating labortories had a consistent diagnostic accuracy of >98%; confirming that PCR–if well performed–is a very powerful diagnostic approach. These results confirm the necessity of EQAS to ensure proper performance of NAATs for the detection of STH and SCH in stool.

In stool samples most false-negative results were reported for the detection of *T. trichiura* (29%). The median reported Cq-value for all positive *T. trichiura* stool samples was higher than those of stool samples positive for the other helminth targets (**Table 1**). Although it cannot be excluded that this difference is simply caused by inclusion of stool samples containing relatively low loads of *T. trichiura*, this difference could (in part) also be caused by the previously reported difficulties in extracting DNA from *T. trichiura* eggs, which have a relatively late embryonic development (and therefore not as many cells and genome copies per egg) and a relatively thick and lysis-resistant eggshell [8, 23]. Participating laboratories that performed best in detection of *T. trichiura* in stool (C, G and K) used relatively rigorous lysis methods (combining beat beating and protease K treatment), which suggests that harsh lysis methods are required for sensitive detection of *T. trichiura* eggs in stool.

For DNA samples most false-negative results were reported for *Ascaris* spp. (8%), and most of these false-negative results were reported for the DNA samples distributed in 2019 (n = 6)

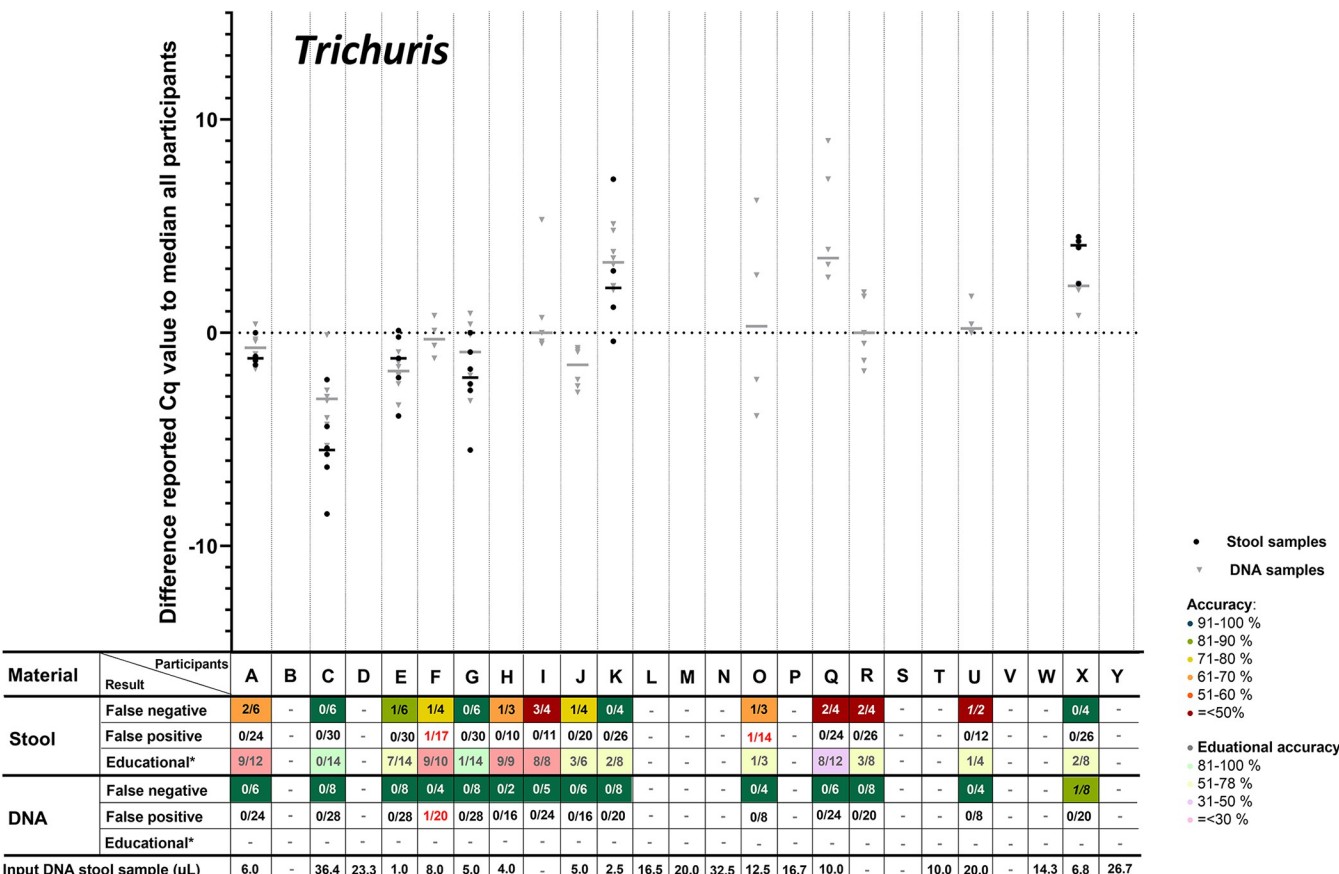

**Fig 5. Difference in reported Cq-values for stool and DNA samples positive for *Trichuris trichiura*.** For detailed explanation of the figure, see legend of Fig 1.

and 2020 (n = 4). Interestingly, most of these participating laboratories detected *Ascaris* spp. in the simultaneously distributed stool samples. The cause for this striking observation was the source from which the DNA sample for *Ascaris* spp. was prepared; adult worm muscle tissue for the 2019 and 2020 samples and from eggs for the samples from 2021 onwards. This source was important as the DNA target for *Ascaris* used by some participating laboratories was the high copy-number repeat sequence specific for germline cells that are only abundantly present in eggs and larvae [24]. Therefore, most of the false-negative results for *A. lumbricoides* in DNA samples appeared to be an artefact of the source of DNA for sample preparation.

As this study showed that the variation in the reported semi-quantitative results (the Cq-values) between laboratories is extremely variable and can be as large as 15 DNA amplification cycles, quantitative PCR results are not comparable between laboratories. This hampers epidemiological surveys and thus potential application for evaluation of efficacy of large scale deworming programs if examined by distinct laboratories that do not use an identical protocol [10, 12, 15]. Hence, harmonization is needed, but given the in this study shown multi-factorial cause of the variation this will be difficult to achieve unless identical protocols and equipment will be used.

Finally, the following limitations of this study should be noted. This study was a retrospective study that analyzed the results of HEMQAS, which was not designed to investigate which NAAT method is most efficient. In addition, given the noted large extent in variations in

NAAT methodology compared to the number of participating laboratories, almost every participant used a unique protocol, which prohibits firm conclusions on what methodology is efficient or not. Distributed samples were selected from a limited number of geographical sites (India, East Africa, Dutch travelers, and immigrants to the Netherlands), which might not be representative for all helminth genome variants around the world. Furthermore, the clinically important helminths *Ancylostoma* spp. and *Schistosoma haematobium* could not be included in this study, as until now no stool (*Ancylostoma* spp) or urine (*S. haematobium*) samples were distributed that contained these helminths. Given the clinical importance of female and male genital schistosomiasis caused by *S. haematobium* [25, 26], future inclusion of this helminth in EQA schemes is necessary. The selection criteria for inclusion of participant results for analysis of correlations between qualitative and quantitative results, resulted in a selection bias for laboratories based in high income countries as those participated for longer periods to HEMQAS and were able to report HEMQAS reports during the COVID-19 outbreak. In contrast to laboratories in high-income countries, laboratories from endemic countries were unable to participate as consistently in EQA schemes. Since a detailed overview of the performance of NAAT in laboratories based in endemic areas in low- and middle-income countries is lacking, this needs further investigation and requires further low-cost EQA implementation in these countries.

## Supporting information

**S1 Data. False positives.**
(XLSX)

**S2 Data. False negatives.**
(XLSX)

## Acknowledgments

The authors express their gratitude to the participating laboratories of HEMQAS and to Nicolette van der Ham (Erasmus MC, Rotterdam), Malathi Manuel (CMC, Vellore), Francesca Azzato (Peter Doherty Institute for Infection and Immunity, Melbourne), and Eric Brienen (Leiden University Medical Center) for expert technical assistance.

## Author Contributions

**Conceptualization:** Annemiek H. J. Schutte, Rob Koelewijn, Jaap J. van Hellemond.

**Data curation:** Annemiek H. J. Schutte, Rob Koelewijn.

**Formal analysis:** Annemiek H. J. Schutte, Rob Koelewijn.

**Funding acquisition:** Jaap J. van Hellemond.

**Resources:** Sitara S. R. Ajjampur, Bruno Levecke, James S. McCarthy, Rojelio Mejia, Steven A. Williams, Jaco J. Verweij.

**Supervision:** Jaap J. van Hellemond.

**Validation:** Sitara S. R. Ajjampur, Bruno Levecke, James S. McCarthy, Rojelio Mejia, Steven A. Williams, Jaco J. Verweij.

**Writing – original draft:** Annemiek H. J. Schutte.

**Writing – review & editing:** Annemiek H. J. Schutte, Rob Koelewijn, Sitara S. R. Ajjampur, Bruno Levecke, James S. McCarthy, Rojelio Mejia, Steven A. Williams, Jaco J. Verweij, Lisette van Lieshout, Jaap J. van Hellemond.

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
