## [Decision Letter · Decision Letter 0]

29 May 2024

Dear Dr. van Hellemond,

Thank you very much for submitting your manuscript "Detection of soil transmitted helminths and Schistosoma spp. by nucleic acid amplification test: results of the first 5 years of the only international external quality assessment scheme." for consideration at PLOS Neglected Tropical Diseases. As with all papers reviewed by the journal, your manuscript was reviewed by members of the editorial board and by several independent reviewers. The reviewers appreciated the attention to an important topic. Based on the reviews, we are likely to accept this manuscript for publication, providing that you modify the manuscript according to the review recommendations. 

Sincerely,

Francesca Tamarozzi

Section Editor

Reviewer's Responses to Questions

**Key Review Criteria Required for Acceptance?**

**Methods**

-Are the objectives of the study clearly articulated with a clear testable hypothesis stated?

-Is the study design appropriate to address the stated objectives?

-Is the population clearly described and appropriate for the hypothesis being tested?

-Is the sample size sufficient to ensure adequate power to address the hypothesis being tested?

-Were correct statistical analysis used to support conclusions?

-Are there concerns about ethical or regulatory requirements being met?

Reviewer #1: Why was the scheme designed for only 1 challenge per year? How does this align with recipient requirements for PT frequency?

Reviewer #2: Line 131: Insufficient information is given about the samples distributed. How were they "Well validated" and by whom? What were the egg counts? Were the stools fresh or frozen? Were they sent out "dry" or in PCR extraction buffer, for example? How were the DNA samples sent out? Purified DNA or DNA seeded into negative stools? What was the quantity of DNA sent out? Was the same quantity sent out each time?

**Results**

-Does the analysis presented match the analysis plan?

-Are the results clearly and completely presented?

-Are the figures (Tables, Images) of sufficient quality for clarity?

Reviewer #1: OK

Reviewer #2: Line 190: Were all the stools positive by light microscopy? if so, PCR performance was poor.

**Conclusions**

-Are the conclusions supported by the data presented?

-Are the limitations of analysis clearly described?

-Do the authors discuss how these data can be helpful to advance our understanding of the topic under study?

-Is public health relevance addressed?

Reviewer #1: OK

Reviewer #2: Discussion line 298: Please rephrase this line; should read positive for....

Acknowledgements: Those named as giving expert technical assistance will have done a substantial amount of work. Please consider whether they meet the criteria for full authorship.

**Editorial and Data Presentation Modifications?**

Reviewer #1: OK

Reviewer #2: Minor grammatical edits needed.

**Summary and General Comments**

Reviewer #1: This was nicely presented and important work.

Reviewer #2: This is a very useful study which highlights the challenges facing EQA in parasitology

PLOS authors have the option to publish the peer review history of their article (what does this mean?). If published, this will include your full peer review and any attached files.

Reviewer #1: No

Reviewer #2: No

Figure Files:

Data Requirements:

Reproducibility:

References

---

## [Editor Report · Decision Letter 1]

25 Jul 2024

Dear Dr. van Hellemond,

We are pleased to inform you that your manuscript 'Detection of soil-transmitted helminths and Schistosoma spp. by nucleic acid amplification test: results of the first 5 years of the only international external quality assessment scheme.' has been provisionally accepted for publication in PLOS Neglected Tropical Diseases.

Best regards,

Jong-Yil Chai

Section Editor

Francesca Tamarozzi

Section Editor

The revised manuscript is now acceptable by PLoS NTD.

---

## [Editor Report · Acceptance letter]

4 Aug 2024

Dear Dr. van Hellemond,

We are delighted to inform you that your manuscript, "Detection of soil-transmitted helminths and Schistosoma spp. by nucleic acid amplification test: results of the first 5 years of the only international external quality assessment scheme.," has been formally accepted for publication in PLOS Neglected Tropical Diseases.

Best regards,

Shaden Kamhawi

co-Editor-in-Chief

Paul Brindley

co-Editor-in-Chief
